# Needs and capabilities for improving poultry production and health management in Indonesia

**Lorraine Chapot**[1,2]*, **Rebecca Hibbard**[1,2], **Kurnia Bagus Ariyanto**[3], **Kusnul Yuli Maulana**[3], **Havan Yusuf**[3], **Widya Febriyani**[3], **Angus Cameron**[1], **Mathilde Paul**[2], **Timothée Vergne**[2], **Céline Faverjon**[1]

1 Ausvet Europe, Lyon, France, 2 IHAP, INRAE, ENVT, Université de Toulouse, Toulouse, France, 3 Ausvet Representative Office Indonesia, Jakarta, Indonesia

* lorraine.chapot@ausvet.com.au

**Data Availability Statement:** Interview records and transcripts are not publicly available due to privacy and ethical restrictions, as they contain personal and contextual information that may lead

## Abstract

In Indonesia, the development of the poultry industry is facing numerous challenges. Major constraints include high disease burdens, large fluctuations in farm input and output prices, and inadequate biosecurity. Timely and reliable information about animal production and health can help stakeholders at all levels of the value chain make appropriate management decisions to optimize their profitability and productivity while reducing risks to public health. This study aimed to describe the challenges in the Indonesian poultry industry, assess stakeholders' needs and capabilities in terms of generating and using poultry information for making production and health management decisions, and identify levers for improvement. Interviews were conducted with a diversity of key informants and value chain actors in five Indonesian provinces. Thematic analysis was applied with an interpretivist approach to gain an in-depth understanding of the lived experiences of various stakeholders and their opinions as to what might constitute appropriate solutions. Our findings indicate that market and political instability, ineffective management of poultry data, and limited inter-sectoral collaboration are limiting the development of the sector. Increased intersectoral cooperation is needed to implement standards for data collection and sharing across the industry, provide education and practical training on the use of information technologies for farm management, and accelerate research and innovation. Our study can contribute to the development of data-driven tools to support evidence-based decision-making at all levels of the poultry system.

## Introduction

As an accessible and affordable form of livestock, poultry production is an important source of nutrition and income in many low- and middle-income countries (LMIC) [1]. In Indonesia, it is one of the fastest-growing agricultural subsectors and is expected to play a key role in addressing the growing demand for animal protein [1–4]. However, rapid intensification and

to the identification of the interviewees. Data is stored in Ausvet's internal cloud-based storage system. Access to anonymized and de-identified data may be requested by contacting Ausvet Europe (contact@ausvet.eu) and the Medical and Health Research Ethics Committee of Gadjah Mada University, Indonesia (komisietik@ugm.ac.id).

**Funding:** This study was conducted as part of the TRANSFORM (Transformational Strategies for Farm Output Risk Mitigation) funded by USAID (Cooperative Agreement No. 7200AA21CA00004). The funders had no role in study design, data collection and analysis, decision to publish, or preparation of the manuscript.

**Competing interests:** The authors have declared that no competing interests exist.

globalization come with high public health risks. Furthermore, although vertically integrated players dominate the market with 60% of production coming from industrialized farms [3], the poultry sector still involves a large number of small and medium enterprises operating in low biosecurity settings, which creates risks for disease emergence and spread [2, 5]. High-impact endemic diseases such as Newcastle disease (ND) and highly pathogenic avian influenza (HPAI) continue to cause major economic losses [5, 6]. Barriers to maximizing the potential of poultry production also include farm input and output price fluctuations, limited resources, lack of education on biosecurity and management, and inadequate market regulation [3, 5, 7].

These constraints have the potential to be addressed by providing access to timely and reliable information on poultry production and health through technical innovations [8–11]. Although many individual actors and organizations in Indonesia currently use poultry-related data to improve their business efficiency, their specific information needs, analytical capacities, and willingness to share their own data vary greatly. In the absence of industry-level standardized mechanisms for collecting and managing data, integration and interpretation remains challenging. In particular, the existing National Animal Health and Production Information System iSIKHNAS (*Sistem Informasi Kesehatan Hewan NASional*) has had limited uptake in the poultry sector [12]. Failure of new interventions or technologies to achieve the desired impact despite effective delivery is often attributed to a lack of consideration of the participants' needs [13–16]. Therefore, assessing the needs, capacities, and priorities of the people whose behaviour will dictate the success or failure of management strategies is a crucial step in the design of effective and sustainable interventions [13, 16, 17].

There is growing recognition of the value of qualitative approaches for facilitating the design and implementation of animal health interventions or innovations. Qualitative approaches can offer valuable insight into the views and perceptions of individuals or groups and provide an in-depth understanding of the social, cultural, and economic factors that shape the context in which they are situated [16, 18]. Thematic analysis (TA) is a theoretically flexible method that can be applied to a diversity of contexts and research questions that has been widely used in qualitative research to provide rich and detailed accounts of qualitative data [19–22]. Building on previous analysis of the Indonesian commercial layer value chain and information networks [12], this study uses TA to explore the challenges and priorities of actors of the commercial layer sector regarding production and health management and assess their needs and capacities for informed decision-making. The specific topic of antimicrobial use is the subject of a specific study and is therefore not addressed here [23]. The research questions were formulated as follows: 1) What are the challenges faced by stakeholders of the Indonesian commercial poultry production system and how are these challenges prioritized? 2) What are stakeholders' current practices, needs, and capabilities in terms of collecting and using poultry health and production information to make health and production management decisions? and 3) What are their perspectives regarding potential solutions to address current gaps in the decision-making process? The findings will help researchers and policymakers to design effective and acceptable innovations to support the development of the industry in a sustainable way.

## Material & methods

This paper is reported according to the Standards for Reporting Qualitative Research (SRQR) [18].

### Qualitative approach and theoretical assumptions

The use of a qualitative approach encodes particular ontological and epistemological assumptions around knowledge creation that drive the selection of matching methodology [21, 24].

This study was conducted within an interpretivist (experiential) paradigm in order to prioritize the "empathic understanding" of participants' lived experiences and the meaning they ascribe to them [25]. Interpretivism acknowledges that different people experience and understand the same "social reality" in different ways, and therefore seek to extract knowledge from the subjective interpretations of the people who experience it [25, 26]. This approach was consistent with the study's objectives of emphasizing poultry stakeholders' subjective accounts of their experiences and interpretations.

**Sampling and study area.** Sampling was purposive to maximize diversity and prioritize information-rich individuals from the Indonesian poultry sector. Initial participants were identified through consultation with local collaborators and policymakers based on their knowledge or strategic role in the poultry sector, and subsequent participants were recruited through snowball sampling until we felt we had reached a sufficient diversity of views (i.e., actors operating at different levels of the value chains and geographic places) and that data was sufficiently rich to answer the research question. The final sample included policymakers from the Directorate of Livestock and Animal Health Services (DGLAHS), representatives from non-governmental organizations, industry associations, university professors, technical service officers (TSO) from input companies, poultry company managers, and farmers. Although the layer sector was primarily targeted upon the government's request, several interviewees also had experience in the broiler sector. The study area covered five provinces within which most layer poultry production is concentrated: DKI Jakarta, Yogyakarta, Central Java, East Java and West Java.

**Data collection.** Semi-structured interviews were used to collect rich, descriptive and contextually situated data for exploring the range of attitudes, experiences, perceptions, and behaviours of a diversity of poultry stakeholders. This flexible format allowed adjustment of the emphasis and order of the topics according to the interviewee's responses and for interviewers to follow-up more extensively on any unexpected themes that emerged during the discussion [27–30].

To build *a priori* semi-structured interview guides, an initial literature review was conducted. Based on this review, the following topics were included: (1) Role and challenges of the interviewee in their field of work; (2) Practices and challenges related to the prevention and control of poultry diseases; (3) Antimicrobial use and stewardship, and (4) Practices and challenges related to the management of poultry-related information. The interview guide is provided in S1 Appendix. The questionnaire was reviewed and practiced amongst researchers and piloted with an independent small-scale farmer to ensure understandability. Interviews were conducted by two separate teams composed of one English-speaking researcher, two Indonesian-speaking researchers, and one professional translator with experience in animal health. Interviews were conducted at the interviewee's workplace, in public spaces such as cafes or restaurants, or online depending on logistical constraints. Representatives from associations, institutions, and government bodies were interviewed in groups of two to five people.

**Data analysis.** Thematic analysis of interview transcripts was conducted following the framework from Braun and Clarke [22] and taking into account recommendations from Castleberry and Nolen for increasing consistency in coding and ensuring transparency and thoroughness of the analytical process [31]. Although this framework is described linearly, several cycles of coding and interpretation were performed iteratively throughout the analytical process. A codebook was developed to map the coding process [32] and ensure consistency across transcripts. It also aimed to demonstrate rigour in the methodology by providing a clear audit trail [33]. The software NVivo [34] was used to facilitate data management.

1. *Data familiarization*. Interviews were audio recorded and transcribed verbatim in the original language (either Indonesian or English). English recordings were transcribed by English-speaking members of the research team and Indonesian recordings were transcribed and translated by Indonesian native speakers. All verbatim transcripts were reviewed carefully by the Indonesian-speaking members of the research team before and after translation to assess quality and become familiar with the data. Where necessary, unclear data excerpts were checked against audio recordings or field notes and annotated with contextual explanations to inform the interpretation.

2. *Initial coding*. The entire dataset was coded systematically using a primarily inductive or "data-driven" approach [35, 36] (i.e., without committing to a pre-existing theory or framework) to best represent meaning as communicated by participants. Deductive coding was used to some degree to ensure meanings were relevant to the research question and provide more details on some aspects of interest. This combined approach is appropriate for studies that aim to explore specific issues but also leave scope for unexpected aspects of participants' experiences or interpretations to be brought up [37, 38]. Semantic (explicit) and latent (implicit) codes were developed to allow the data to be both described and interpreted [22]. An initial codebook was developed, including the following components: label, definition, and ideas to include or exclude. When coding the excerpts from interview transcripts, the researcher retained as much contextual information as needed to ensure quotes would be understandable.

3. *Generating themes*. Once all data had been coded, the data was reviewed to determine how codes may be grouped to form themes or sub-themes [35]. "Keyness" or meaningfulness was the central criterion for determining themes (i.e., whether it captures something important in relation to the research question) [22, 31].

4. *Reviewing themes*. Themes were reviewed against the entire dataset to determine whether they formed a coherent pattern and were relevant to the research question [22, 38, 39]. Codes or themes that shared the same underlying meaning were merged, while those that did not fit within the overall analysis were discarded and their data re-coded as appropriate to facilitate the most meaningful interpretation of the data. The first author recorded all decisions to rename, merge, discard, and sort codes into themes in NVivo to keep track of the changes and allow for going back and re-code previously coded material to ensure consistency in coding.

5. *Refining and defining themes*. Themes were reviewed individually and in relation to one another. An accompanying narrative was written for each theme and sub-theme.

6. *Producing the report*. Themes were organized together with their analytic narratives to illustrate the overall story in response to the research question. Direct quotations were collated in a table (S2 Appendix). All quotations were referred to in the main text using identifiers and the most relevant were reproduced in full to demonstrate how themes are rooted in empirical data and allow the reader to assess the validity of the interpretation. Minor grammatical and linguistic corrections were made by the researcher to ensure comprehension. Where necessary, contextual explanations were added in brackets.

### Ethics statement

This study obtained ethical approval from the Medical and Health Research Ethics Committee from the Faculty of Medicine, Public Health and Nursing of Gadjah Mada University (Ref. No.

KE/FK/0675/EC/2022). The interview started with a self-introduction from both the interviewer and interviewee, following which the interviewer explained the scope, objectives, and interview process. Verbal consent to participate in the study and record the interview was obtained from all interviewees in the presence of all research team members and documented using a recording device. Data was stored securely in restricted access folders. Transcripts were relabelled with code names before analysis and any information that could lead to the direct identification of the interviewee was removed from the quotes. At the end of the interview, interviewees received a non-monetary gift for their participation.

## Results

A total of 34 individual and 7 group interviews involving 56 participants were conducted from 27 June to 22 July 2022 with value chain stakeholders, experts and policymakers, of which 38 were audio recorded. Table 1 presents the number of interviews according to stakeholder category. Although the layer sector was primarily targeted upon the Indonesian government's request, many interviewees provided information or had a personal experience in the broiler sector.

Five major themes were developed including 16 sub-themes which are detailed in the codebook (S3 Appendix). The accompanying narratives are described below.

### Theme 1: An increasingly complex and uncertain business environment

**Economic vulnerability of poultry producers.**   Poultry producers generally were concerned about high production costs in Indonesia which they told us greatly limited their profit margins. Most small- and medium-scale independent farmers experienced economic hardship and appeared to be the most vulnerable to changing market circumstances due to insufficient

**Table 1. List of interviews.**

| Stakeholder category | Number of interviewees |
|---|---|
| Representatives of integrated poultry companies | 6 |
| Company manager | 4 |
| Company veterinarian | 2 |
| Small and medium layer farmers | 12 |
| Representatives of industry associations * | 4 |
| Poultry farmer association | 2 |
| Pharmaceutical company association | 1 |
| Veterinarian association | 1 |
| Government agents * | 4 |
| Representatives of research & development organizations | 5 |
| University professor | 4 |
| Research institution rep. | 2 |
| Non-Governmental Organization | 1 |
| Representative of a feed company | 1 |
| Representative of a pharmaceutical company | 1 |
| Technical service officers (TSO) | 5 |
| Pharmaceutical company officer | 4 |
| Feed company officer | 1 |
| Poultry shop owner | 1 |

* Group interviews

cash flow or poor financial management. Farmers felt this was most apparent during the COVID-19 pandemic, which led to a sharp decrease in the number of independent smallholders, while contract farmers' losses were absorbed by their companies. Farmers generally expressed a pessimistic view regarding the future and stressed the psychological burden of maintaining their businesses in the face of competition and uncertainty (a1, a2, a3, a4).

*"Only big integrators, big companies can survive because they have a lot of money. But small farmers, many of them they close, they sell their cages [. . .]. It's quite challenging to be a farmer in Indonesia."*

(a4)

**An unstable and unpredictable market.** A lack of coordination between supply and demand appeared to be a recurring issue, causing frequent and significant egg and DOC price fluctuations that could not be predicted based on foreseeable geographical and seasonal variations. Farmers felt they had no control over prices and no choice but to follow market trends, without any external support (a5, a8).

*"That's why the breeders now are like a tossed ship. Like the ship in the middle of the sea doesn't know where it's going, its purpose is not clear. We must survive alone, we must fight alone."*

(a5)

Interviewees told us that the lack of price regulation and official certification schemes opened the door to uncontrolled production and fraudulent practices, allowing, for instance, farmers or traders to artificially increase prices by retaining eggs or by using false labels (e.g., for omega eggs) (a6, a7).

**Difficulties in sourcing farm input.** Price and availability of day-old chicks (DOCs) and poultry feed, which together account for more than 80% of farmers' production costs, were the main concerns of most poultry producers. Indonesia's heavy reliance on imports for the supply of DOCs and feed raw materials (e.g., meat-bone-meal, soy-bean-meal, corn, etc.) made local production highly sensitive to international commodity prices, exchange rate variations, and global shocks such as the COVID-19 pandemic or the Ukraine-Russia war (a8). We were also told that government import policies and subsidy schemes also had a strong impact on availability and price stability. The Directorate General of Livestock Farming and Animal Health has established import quotas on grandparent DOC in an attempt to achieve a balance between supply and demand. However, interviewees told us that the quotas appeared to be determined with limited stakeholder consultation and based on inaccurate data, leading to an imbalance between supply and needs (a9).

*"The government imports this GPS* [grandparent stock] *as if they are doing whatever they want. This means that the need is not calculated, how much is needed for DOC [day-old chicks], how many eggs are needed, oh if I have to import that much, there won't be any."*

(a9)

Farmers believed that the government's strategy of favouring local corn production by limiting imports has created many difficulties for sourcing poultry feed. In their opinion, local corn production remains insufficient and sub-optimally managed (a10), with temporal and geographical imbalances between corn and poultry production coupled with logistical issues in

conveying feed raw materials. Farmers also reported difficulties in accessing subsidized corn and complained of its poor quality and prices being higher than unsubsidized corn. This led feed dealers and farmers to develop variable coping strategies, which negatively impacted the quality of feed (e.g., replacement with wheat, use of fake meat-bone-meal, mixing with sand).

**Unequal power dynamics.**   As Indonesian poultry production has intensified and as integrated companies have developed, farmers felt that integrators' monopoly in the domestic market was being reinforced (a11). Independent farmers were becoming increasingly dependent on large integrated companies for the supply of production input and had to comply with their terms and conditions, such as the bundling of feed and DOCs. Many independent farmers expressed a fatalistic view, believing that they did not have the capacity to compete and were doomed to be overtaken by large players (a12, a13, a14).

> "It is impossible for us small farmers to face them. They have DOC [day old chicks], they have feed, they have medicine, they have vaccines, even the last one, they have all the final results. [. . .] My question is, will integrators replace us after we run out?"

(a12)

## Theme 2: Limited collaboration among government, academia, and industry stakeholders

**Government perceived to have weak interest and involvement in the poultry sector.**
From the perspective of private stakeholders, the government had little interest in poultry and focused more on cattle. In particular, small-medium independent farmers felt abandoned by the government and complained of a lack of financial support (b1, b2).

> "Our association [of small poultry farmers] still needs to improve our relationship with the government because government policies often do not suit our needs. Of course, the government must consider the interests of prominent business people, but this policy is ultimately not in the interests of our farmers."

(b2)

Being more advanced in terms of knowledge and technology, large-scale farms or companies failed to see the benefits of interacting with the government and often displayed a distrustful attitude towards them (b3, b4, b5). Interactions were generally limited to the NKV (*Nomor Kontrol Veteriner*, veterinary control number) certification or compartmentalization program for Avian Influenza (AI) (b6).

**Information gaps lead to inadequate management of poultry diseases.**   Private actors were generally reticent about sharing their internal data and rarely reported diseases to authorities for fear of penalties (b7). The official animal information system iSIKHNAS, which has been successful in cattle, was perceived by interviewees as ineffective in poultry (b8). Therefore, data held by the government was considered incomplete or inaccurate, leading to inadequate policy-making and response to industry needs, and in particular late or no response to disease outbreaks (b9, b10, b11).

> "After all, the government is often not ready, especially when facing poultry disease outbreaks. [. . .] If we weren't prepared, we would have collapsed a long time ago for various reasons."

(b10)

Many mentioned the first AI crisis in 2003, telling us that the government failed to recognize the threat, forcing farmers to manage the issue themselves by sourcing vaccines from China outside any legal framework (b11, b12).

*"the government at that time* [during the first AI outbreak in 2003]*, they were unwilling to declare that we had been infected by avian influenza. [. . .] The ability of our government to address this issue was very weak."*

(b11)

**Industry actors perceive academic education and research as disconnected from real-life.** Actors from the private sector felt that the academic world was disconnected from the reality in the field. A farming businessman suggested that poultry sciences should be taught by private industry experts to ensure knowledge is rooted in real life and to better prepare future practitioners (b13). However, he deplored academia's closed attitude towards external lecturers (b14).

*"Academics are usually born in a community where a system has been formed. [. . .] They feel confident and do not need advice from others. [. . .] That's their weakness. They have a good brain but can't accommodate outside advice"*

(b14)

Additionally, he considered academics' way of communicating to be inadequate for farmers and focused too much on the theory compared to the practicalities (b15).

## Theme 3: Inadequate on-farm management of poultry production and health

**Multiple and increasingly complex health challenges.** Avian Influenza, Newcastle disease (ND) and colibacillosis were the most frequent diseases mentioned by farmers. Other significant health hazards mentioned included chronic respiratory disease (CRD), coryza, infectious bronchitis (IB), infectious bursal disease (IBD or Gumboro), and general metabolic diseases due to environmental factors. According to farmers, although health authorities recognized AI and ND as priorities, they tended to neglect non-zoonotic diseases that are still responsible for important economic losses such as IB or CRD (c1). While many farmers had experienced devastating AI outbreaks in the past, they reported a reduction in their occurrence and severity following the wide use of vaccination and improvement of biosecurity (c2). However, they also described a shift in disease symptoms, from high mortality to more unclear signs, making it increasingly difficult to diagnose (c3, c4).

*"It is different from 10 years ago [. . .] In terms of disease, nowadays cases of disease are increasingly complex, they do not stand alone, so the symptoms are not clear and this does require a lab approach"*

(c3)

Interviewees also pointed out the role of environmental risk factors in disease introduction and spread, including high poultry population and farm densities, low feed and water quality, and exposure to extreme weather events.

**Lacking or inadequate management of poultry health.** According to stakeholders responsible for providing support to farmers (including research institutions, NGOs, university professors, and technical service officers), many small-scale independent farmers lack the necessary knowledge and resources to implement adequate biosecurity and health management, and absence of biosecurity practices remain frequent (e.g., mixing of age groups, no all in-all out) (c5). Diseases are generally managed by farmers themselves based on their own experience, without seeking laboratory diagnosis. In addition, farmers thought that prescription and use of medicines were driven by the commercial interests of private veterinarians and technical officers from pharmaceutical companies (c6). With the rapid growth of the sector, independent farmers and company managers mentioned struggling to hire qualified poultry workers with sufficient awareness and knowledge about biosecurity and farm management (c7, c8). Interviewees from research and non-governmental institutions also highlighted the responsibility of the government in these issues:

*"The problem is that for animal health, if you look at the budget here in the Ministry, the budget has been going down year after year, livestock production is much more important than animal health, so it's partly the problem of the government itself"*

(c9)

Control and support from health authorities to farmers appeared to be lacking due to insufficient financial and human resources, but also due to the unstable political and administrative environment which farmers told us frequently disrupted the funding and implementation of national health programs (c9, c10). This was even more critical at the time of the study, as the government had reallocated most of its resources to the management of a nationwide FMD outbreak.

**Resistance to behavioural changes.** Several farmers and technical service officers described a generational gap regarding farming practices. They mentioned it was difficult to educate the old generation of farmers and update their knowledge about poultry farming because of their unwillingness to change practices that have been passed down through generations (c11, c12).

*Because they feel right, it is difficult to change the mindset of farmers. [. . .] And sometimes that's wrong, because they have the principle of "I've been there for decades, what are you new kids doing?", it's sometimes difficult for us to go there for education"*

(c11)

In contrast, the younger generation was thought to be more aware of the importance of biosecurity and more willing to use IT tools for farm management.

## Theme 4: Insufficient capacity to collect and use poultry health and production data

**Capacity to collect and analyze farm data varies significantly among stakeholders.** Most of the small- and medium-scale independent farmers we interviewed collected farm data on a routine basis using a combination of paper-based and spreadsheet systems. Nonetheless, some of them felt they lacked the necessary technical skills and resources to collect data comprehensively and analyze it to its full extent (d1, d2). Although integrated companies and associations can manage large amounts of data, their internal system appeared to operate in silos

on an individual farm or customer basis, thereby limiting their analytical capacity (d3, d4). Management of data by the government itself was also thought to be suboptimal by some participants, as highlighted by an industry association that criticized what they described as burdensome submission procedures and underutilization of data (d5).

> *"Even with the government, we feel that they might not be able to provide valid data. [. . .] The government needs to be encouraged the importance of actually compiling this data so it can actually be used for something. It's not just asking for data and then stop there, because they don't have any system to compile them."*

(d5)

Additionally, issues of quality, representativeness, and timeliness of data collection were considered by both public and private stakeholders to impair their ability to forecast trends and prepare for future shocks. Many interviewees expressed doubts regarding the validity of public data, stressing their belief in its lack of representativeness and potential falsification, especially for population data for which there is an incentive for farmers to underreport to avoid taxation (d5, d6, d7).

**Dissemination of information relies mostly on informal channels.** Information-sharing by interviewees relied largely on informal channels such as WhatsApp (WA), SMS, or personal interaction during stakeholder forums or industry events (e.g., seminars, professional fairs). Most interviewed farmers described being part of regional WA groups through which they share information about prices, disease events, and seek advice from fellow farmers or experts. Likewise, companies used WA to facilitate communication amongst their technical staff, collect farm data, and provide technical support to workers or customers (e.g., support for diagnosis). Because of their close relationship with farmers, technical service officers from input companies had a central role in the transmission of information, acting as bridges between separate business networks (d8).

> *"Information spreads through suppliers. [. . .] If there's a big problem, the supplier usually has a stake. For example, if there is a virus, later, the vaccine supplier will announce to Blitar farmers through a seminar, "So we got information that this virus has started to enter Blitar." [. . .] However, between farmers, no one wants to tell each other."*

(d9)

Although the farmers we interviewed remained reluctant to directly disclose information to other farmers or government agents, they told us they were more open to discussing with visiting officers who subsequently spread the information within their customer network or reported to their company to inform business decisions (e.g., production of specific vaccines) (d8, d9, d10).

**Barriers to data sharing.** The farmers and other industry stakeholders interviewed were generally reluctant to share their internal information, especially disease events and performance data. Interviewees stressed the importance of personal prestige but also the fear that disclosure of internal information would bring negative consequences, such as the loss of competitive advantage, close-down of business, trade disruptions, and increased taxation (d11, d12, d13). Private stakeholders often did not see any benefits of reporting health issues, especially large companies that felt they had sufficient resources to manage them internally without external intervention (d14, d15).

*"it is not easy to get the data [from sector 3 farmers], they don't want to share any information, especially if there is no benefit for them. [. . .] the key is I think, how to show and to prove they get the benefit, and how to prove this really secure and confidential."*

(d15)

Poultry health and production data was therefore managed through multiple systems with different standards and procedures, which resulted in high variability and discrepancies between private and official data, further impairing its usability.

## Theme 5: Leverage points for development

**Strengthening intersectoral collaboration.** Despite most actors working in silos, many recognized the need for increased collaboration. Farmers were increasingly developing into cooperatives or partnerships to strengthen their competitiveness against larger players and facilitate access to feed and subsidies (e1, e2). The small- and medium-scale independent farmers in particular appeared to be those most in need of peer support, as they did not benefit from government subsidies (e3). Alternatively, an increasing number of farmers told us they are converting to the nucleus-plasma system (i.e., contracting with a company) which is believed to offer better technical support and financial security (e4).

Public and private stakeholders generally appeared to adopt a distrustful attitude towards each other and were reluctant to cooperate, as stressed by both a government agent (e5) and a pharmaceutical company (e6).

*"If farms really think the government is corrupt, then they also feel the government is not independent [. . .] if you want to use the Public Private Partnership approach, the main problem is information confidentiality and good governance."*

(e5)

*"It would be better if the government could support us. [. . .] If they collaborated with us, they could work faster because we have the laboratories so we could help them. But I think they don't trust us yet."*

(e6)

Representatives from associations such as ADHPI (*Asosiasi Dokter Hewan Perunggasan Indonesia*, association of poultry veterinarians), PINTAR (local association of poultry farmers in Lampung), or ASOHI (*Asosiasi Obat Hewan Indonesia*, Indonesian Veterinary Medicine Association) stressed their role as intermediaries in facilitating communication and building trust between private and public actors. For instance, ADHPI organizes a multi-stakeholder forum called OBRASS (*Obrolan Santai Seputar Unggas*), to provide opportunities for informal discussion between private and public stakeholders. They also told us they contributed to policymaking by conveying stakeholders' issues to the government and supporting the dissemination and implementation of official guidelines and regulations.

Several associations and private companies highlighted the advantages of collaborating with universities and institutions for research and development (e7), for instance by partnering with local farmers to develop and evaluate new feed additives. However, they felt there was a lack of communication from institutions regarding current research (e8).

**Addressing education and research gaps.** Associations, input companies, and integrators told us they were investing significant efforts to educate farmers on farm and poultry health management to help them increase their productivity and profitability (e9). Education was

provided through seminars led by experts from companies or universities, onsite training, or newsletters. However, those providing the education felt that important knowledge gaps remained, especially in smallholders. Farmers generally relied on their practical experience and a diversity of informal information sources including technical service officers, university professors, the internet, fellow farmers, and veterinarians who may sometimes lack scientific updates (e10). A local entrepreneur thought training should be tailored to the specific needs of small-scale farmers (e11):

> *"These past few years, I have tried to educate middle to lower-level farmers whose animal health awareness still needs to improve. [. . .] However, my efforts are still very far away. [. . .] The seminars they need are about disease management and production management in a language they can easily digest"*

(e11)

There is also a strong need for research and innovation in poultry nutrition and health, especially for alternatives to antimicrobials and raw feed materials that can be locally produced (e12). However, according to academics, research in poultry is not yet sufficiently developed to support the rapid growth of the industry (e13).

> *"Even though we are number four in Asia [. . .] it is not easy to find funding for research in poultry. [. . .] And indeed, that is our weakness, even though the poultry industry in Indonesia is quite large, the research is not yet developed."*

(e13)

**Improving data integration and real-time surveillance.** Stakeholders were increasingly convinced of the importance of good monitoring of production and health data as key to successful business development (e14, e15). Companies and a significant proportion of independent farmers used production data to evaluate performance, provide early warnings, and guide disease investigation. However, the level of analysis was variable and several interviewees felt they lacked the capacity to analyze it to its full potential. This trend is also driven by evolutions in market demand and regulations. For instance, access to international and high-value markets is dependent on providing internal surveillance data to demonstrate AI-free status (e16).

Nonetheless, important data gaps remain to be addressed. Independent farmers are particularly in need of commodity price data to manage their sales and purchases, while feed and pharmaceutical companies are most interested in health information to provide better services to their customers (e.g., tailored vaccination programs) and plan their production accordingly. Additionally, the lack of data integration and timeliness impede their capacity to anticipate and respond to potential future threats (e17, e18), as emphasized by a pharmaceutical company:

> *"Now we only know when the disease is here. But the disease is not just there all of a sudden, there is this process, and we need to know this process [. . .] We don't want to fight fires, once we have fires and we fight it, we don't want to do this. We want to know the pattern so that we can be better prepared."*

(e17)

## Discussion

Our study offers insight into how the Indonesian poultry system operates, describes the main issues faced by the stakeholders, and identifies opportunities to address them. Focusing primarily on the layer sector allowed us to fill an important research gap, as most past studies have been conducted in the broiler sector. Our findings also highlight the highly interconnected nature of the Indonesian poultry system and may therefore be of interest to those who seek to understand global poultry production and trade dynamics, especially in the wider SEA region. Finally, our study demonstrates the value of qualitative approaches for exploring complex systems, moving beyond the technical or biological aspects to encompass the broader social, cultural, and economic determinants of health.

### A complex and uncertain environment in which health is not a priority

The main difficulties for both independent farmers and companies were associated with market price instability and insufficient supply of high-quality feed. Inadequate infrastructure and dependence on foreign production meant that prices of farm supply and commodities were subject to large and unpredictable fluctuations, which complicated the planning of production and led to recurrent supply issues. These challenges, which are commonly reported in studies on the Indonesian poultry industry [3, 10, 40, 41], are also widely experienced by many Asian countries such as Vietnam and The Philippines [42–44]. As generally observed in LMICs [5, 44], the small-medium independent farmers appeared to be the most vulnerable, as they often lacked the logistical and financial capacities to absorb rapid changes in their business environment (e.g., by stocking up on feed while it is affordable). In the absence of developmental prospects, they tended to focus on surviving rather than growing. In this context, health management was considered by farmers to be a secondary concern compared to maintaining profitability. Additionally, farmers seem to have become less concerned with diseases that used to have devastating consequences (such as AI and ND) due to the wide use of vaccination and likely risk habituation. This mirrors findings in another study in Indonesia which reported low concern for disease spread in villages that have had suspected or confirmed cases of HPAI and highlighted a disconnection between awareness of HPAI risk and biosecurity practices [45].

### Ineffective management of poultry production and health data

Monitoring and analysis of production and health data is becoming a necessity to maintain competitiveness in a rapidly changing business environment. Making appropriate use of data enables farmers to plan production in the long term, identify seasonal and geographical disease patterns to develop early warning systems, adapt to changing climatic conditions and more generally support daily management decisions [8–10, 46]. For instance, a study has shown that a lack of data prevented Indonesian farmers from responding to climate change risk [46]. In our context, limitations in data availability and analytical skills impeded stakeholders' capacity to make accurate forecasts and prepare for potential threats. Independent farmers in particular were in need of more information on commodity prices, while input companies were interested in obtaining a more comprehensive picture of the disease situation in the field to provide better services to their customers (e.g., specific feed formulation or vaccine programs). Developing data-driven decision-support tools could help all stakeholders access and analyze information to support risk management and daily decision-making [8, 9, 46–48]. As already highlighted in previous studies, the adoption of these technologies is currently limited by the aging of the farmer population and low level of information technology (IT) education [46–49]. However, IT literacy is expected to increase among the younger generation of farmers

[47–49]. Nonetheless, important educational gaps remain, especially for independent smallholders who lack formal training on biosecurity, and farm and financial management. At present, it appears that research and education in poultry sciences are insufficiently developed to fulfil the industry's needs for innovation and skilled graduates. To address part of this issue, a study suggested creating a research data bank that could be used as a reference for farmers and their advisors [46].

## Limited intersectoral collaboration and unequal power dynamics

Our final theme pertained to the current and needed efforts to achieve greater efficiency and information-sharing in the poultry sector. Weaknesses in coordination and communication between government, academia, and the private sector have been identified as a key issue in agricultural development, leading to inefficiencies in value chain operations, inadequate policy-making, siloed information systems, and underutilization of research results [46, 50, 51]. Both public and private sector interviewees from our study stressed the importance of intersectoral collaboration for supporting the growth of the industry. For instance, some large companies advocated for the pooling of resources and the development of joint strategies with the government to effectively address industry-wide threats such as disease outbreaks. At the farmer level, the development of direct transactions with traders, aggregation of small-scale production through cooperative or contract farming, and development of public-private partnerships were suggested as potential solutions to achieve greater efficiency and coordination in value chains and increase smallholders' inclusiveness in high-value markets, in line with recommendations from other studies [44, 52, 53]. Such initiatives have shown results in other Indonesian value chains such as dairy, palm oil or cocoa [54, 55]. Another successful example comes from the Productive Alliance approach supported by the World Bank, which involves the development of horizontal integration among smallholders and the establishment of public-private partnerships. In countries of Latin America, Sub-Saharan Africa, and East Asia where this approach has been implemented, it has notably led to increases in productivity, income, market integration, and inclusion of vulnerable groups [56]. Multi-stakeholder platforms are also seen as a promising means to increase stakeholders' awareness of their different interests, needs, and interdependence and promote better collaboration in agricultural research and development [44]. This could help address the issue of unequal power dynamics and improve the dissemination of research findings and regulations.

## Policy implications and recommendations

The present study identifies three areas of intervention for improving the efficiency and sustainability of poultry production in Indonesia. First, leveraging and reinforcing inter-sectoral collaboration could contribute to better synchronization of supply and demand, coordinate responses to industry-wide threats such as disease outbreaks, and improve smallholders' inclusiveness in modern production chains. Second, integrated and real-time data and surveillance systems would enable early warning and facilitate rapid decision-making in response to new opportunities and challenges. Definition of industry-level data standards would also be beneficial to ensure the interoperability and reliability of data across sectors. Finally, intensified collaboration between academia and the private sector in research and education could generate graduates and innovations to meet the industry's needs, giving particular attention to smallholders whose participation will depend on their capacity to meet evolving biosecurity and production standards. The government also has an important role in ensuring that the institutional and regulatory environment is supportive of these developments. As stressed in a report from the World Bank [54], this would require a complete

understanding of the way the private sector functions, including the traditional sector, with particular attention to the trade of strategic elements such as feed and DOCs. The issues and recommendations identified in this study are well aligned with those formulated by Task Force 6 –Global Health Security and COVID-19 regarding One Health implementation during the 2022 T20 summit in Indonesia [57].

## Limitations of the study and considerations

Special care was taken to include stakeholders at all levels of the value chain and cover a large geographical area, in order to ensure a diversity of views were included in our results, but as we prioritised including those who would provide the most information-rich data, the findings should not be taken as generalizable to all Indonesian poultry farmers. Specific groups which were not included in our study who might have differing perspectives were notably those from the broiler sector and backyard farmers. Further work could examine the specific challenges and needs of these actors. Additionally, gender dynamics were not considered. It is noteworthy that only 12 of the 56 interviewees were female, including only one female farm owner. Previous studies have stressed that the lack of consideration of gender issues can limit the effectiveness of interventions [5]. Exploring the specific challenges and barriers faced by female poultry workers would require further investigation.

Reflexive thematic analysis is a method that capitalizes on the researcher's theoretical assumptions, analytical resources, and skills for generating data, particularly when undertaken within an interpretivist paradigm [31, 33, 58, 59]. TA is not about achieving a consensus in meaning or discovering one "truth" buried in the data, but rather actively interpreting, creating, and telling "stories" with researcher subjectivity understood as a resource to actively co-construct data and results rather than something to be neutralized [20, 59]. This means that the results necessarily include elements of subjectivity, and the procedures and approaches used to ensure scientific rigour and transparency are different from those used in quantitative research or qualitative research with a positivist paradigm. In our case, to ensure transparency, we described in detail the methods and analytical process to make explicit the epistemological position and assumptions that underpin the study's empirical claims and ensure the research process is logical and traceable, as recommended by Castleberry and Nolen [31]. Reflexivity, which aims to critically evaluate how the researcher's own experiences, assumptions, power relationships, and context may have incidentally impacted the research process, was harnessed in the form of reflexive writing and collaborative reflection with participants following guidelines from Olmos-Vega *et al.* [59]. Written documentation of reflections occurring during the research process included field notes, personal memos, and annotations of transcripts to record methodological decisions, researcher-interviewee dynamics, and call attention on aspects of context that may have impacted the data and results. Collaborative reflection was undertaken by re-engaging with participants at the end of each interview to ensure the findings reflected their experiences and identify where interpretations may differ. In addition, findings were reported with attention to Lincoln and Guba's four evaluative criteria for trustworthiness [60]: (1) preliminary findings were triangulated with different researchers' notes and policy documents to build confidence in the "truth" of findings (*credibility*); (2) thick contextual descriptions were provided to enable readers to determine relevance to other contexts (*transferability*); (3) the analytical process was described in detail to ensure consistency and repeatability of findings (*dependability*); and (4) direct quotations from the dataset were included to demonstrate how findings derive from the data (*confirmability*). Although our results are not measurable in a quantitative sense and are not statistically generalizable, they may be considered nonetheless "theoretically generalizable" or transferable, which means they can be

relevant for understanding similar issues in other contexts. The implications of these findings may be used to inform strategies to improve animal health management and guide the design of further quantitative and quantitative studies [61].

## Conclusion

Our qualitative approach allowed us to gain an in-depth understanding of stakeholders' needs and capabilities for managing poultry production and health in Indonesia and highlighted gaps in the decision-making process, as well as potential opportunities for improvement. Market and political instability, ineffective management of poultry data, and limited inter-sectoral collaboration appear to be the main barriers to the development of the sector. Addressing these challenges would require increased cooperation from all sides to develop and implement industry standards for data collection and sharing, provide education and practical training based on real-life situations, and accelerate research and innovation to respond to the evolving market and regulatory requirements.

## Supporting information

**S1 Appendix. Interview protocol (generic).**
(DOCX)

**S2 Appendix. Quotation table.**
(DOCX)

**S3 Appendix. Codebook.**
(DOCX)

## Acknowledgments

The authors thank all the participants who took part in this study and the translators for their valuable support.

## Author Contributions

**Conceptualization:** Lorraine Chapot, Rebecca Hibbard, Kurnia Bagus Ariyanto, Kusnul Yuli Maulana, Havan Yusuf, Widya Febriyani, Angus Cameron, Mathilde Paul, Timothée Vergne, Céline Faverjon.

**Data curation:** Lorraine Chapot, Rebecca Hibbard, Kurnia Bagus Ariyanto, Kusnul Yuli Maulana, Havan Yusuf, Widya Febriyani.

**Formal analysis:** Lorraine Chapot.

**Investigation:** Lorraine Chapot, Rebecca Hibbard, Kurnia Bagus Ariyanto, Kusnul Yuli Maulana, Havan Yusuf, Widya Febriyani, Angus Cameron.

**Methodology:** Lorraine Chapot, Rebecca Hibbard, Kurnia Bagus Ariyanto, Kusnul Yuli Maulana, Havan Yusuf, Widya Febriyani, Angus Cameron, Mathilde Paul, Timothée Vergne, Céline Faverjon.

**Supervision:** Timothée Vergne, Céline Faverjon.

**Writing – original draft:** Lorraine Chapot.

**Writing – review & editing:** Lorraine Chapot, Rebecca Hibbard, Kurnia Bagus Ariyanto, Kusnul Yuli Maulana, Havan Yusuf, Widya Febriyani, Angus Cameron, Mathilde Paul, Timothée Vergne, Céline Faverjon.

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
