## [Decision Letter · Decision Letter 0]

19 Mar 2024

PONE-D-23-43241Needs and capabilities for improving poultry production and health management in IndonesiaPLOS ONE

Dear Dr. Chapot,

Thank you for submitting your manuscript to PLOS ONE. After careful consideration, we feel that it has merit but does not fully meet PLOS ONE’s publication criteria as it currently stands. Therefore, we invite you to submit a revised version of the manuscript that addresses the points raised during the review process.

We look forward to receiving your revised manuscript.

Kind regards,

Biswajit Pal, M.SC., Ph.D

Academic Editor

PLOS ONE

Journal Requirements:

2. Thank you for stating the following financial disclosure: "This study was conducted as part of the TRANSFORM (Transformational Strategies for Farm Output Risk Mitigation) funded by USAID (Cooperative Agreement No. 7200AA21CA00004)."

Additional Editor Comments:

Kindly modify the manuscript as mentioned by the reviewers.

Reviewers' comments:

Reviewer's Responses to Questions

**Comments to the Author**

1. Is the manuscript technically sound, and do the data support the conclusions?

Reviewer #1: Partly

Reviewer #2: Yes

2. Has the statistical analysis been performed appropriately and rigorously? 

Reviewer #1: I Don't Know

Reviewer #2: Yes

3. Have the authors made all data underlying the findings in their manuscript fully available?

Reviewer #1: No

Reviewer #2: Yes

4. Is the manuscript presented in an intelligible fashion and written in standard English?

Reviewer #1: Yes

Reviewer #2: Yes

5. Review Comments to the Author

Reviewer #1: The article is prepared through survey based adoptive research activities, as prepared through qualitative thematic analysis concept. In the article no significant qualitative statistical analysis was observed in the article, which makes it difficult to specify as review or research article. The orientation of the cited references should be verified, if presented following the standard format of the journal.

Reviewer #2: An interesting study, I hope this manuscript will be published and read by all poultry stakeholder

Please see on the manuscript for spesific comment, however a revision is needed.

And if any additional data or information from university/research institution expert, will be broaden the manuscript discussion

6. PLOS authors have the option to publish the peer review history of their article (what does this mean?). If published, this will include your full peer review and any attached files.

Reviewer #1: No

Reviewer #2: No

---

## [Author Response · Author response to Decision Letter 0]

16 Apr 2024

The authors would like to thank the reviewers for their comments and recommendations to improve this paper. Below are the authors’ responses to the points they raised. 

Reviewer 1:

Comment: The article is prepared through survey based adoptive research activities, as prepared through qualitative thematic analysis concept. In the article no significant qualitative statistical analysis was observed in the article, which makes it difficult to specify as review or research article. The orientation of the cited references should be verified, if presented following the standard format of the journal.

Authors’ response: Our study aimed to address a research question that was exploratory in nature – namely to understand stakeholders’ challenges, needs and capabilities related to animal production and health management in the Indonesian poultry sector. For this reason, we used a qualitative approach and inductive analytical method to explore their subjective experiences and the meanings they assign to them. 

The use of qualitative approaches is well established in health sciences (Bannister-Tyrrell and Meiqari, 2020; Faltermaier, 1997), and there are recognised reporting standards for qualitative research articles. In our case, we followed the Standards for Reporting of Qualitative Research (O’Brien et al., 2014). The framework for reflexive thematic analysis we adopted (Braun and Clarke, 2006) has been applied to a variety of research questions and contexts in animal health (Bard et al., 2017; Chomyn et al., 2023; Hennessey and Barnett, 2023). Contrary to content analysis and other semi-quantitative or mixed-method approaches, it does not involve any quantitative measurement (e.g., word frequency counts), but aims at generating results which provide an in-depth, contextualised understanding of the data. By the nature of its scope and scale, qualitative research is not used to generate quantitative estimates of prevalence, correlation between factors or predict outcomes, and is not applied to obtain statistically generalizable conclusions – it therefore does not entail the use of statistical tools. In line with guidance on thematic analysis (Braun and Clarke, 2006), our results were reported in terms of descriptive narratives of the themes, providing an in-depth understanding of the factors that influence decision-making for poultry production and health management from the stakeholders’ perspective. We conducted a number of procedures to ensure rigor and maintain a “reflexive stance” which are described in the discussion lines 720 - 729. A paragraph has been added lines 736 – 741 to better explain what it entails in terms of transferability of the results (Ma, 2000). 

References:

Bannister-Tyrrell, M., Meiqari, L., 2020. Qualitative research in epidemiology: theoretical and methodological perspectives. Ann. Epidemiol. 49, 27–35. https://doi.org/10.1016/j.annepidem.2020.07.008

Bard, A.M., Main, D.C.J., Haase, A.M., Whay, H.R., Roe, E.J., Reyher, K.K., 2017. The future of veterinary communication: Partnership or persuasion? A qualitative investigation of veterinary communication in the pursuit of client behaviour change. PLOS ONE 12, e0171380. https://doi.org/10.1371/journal.pone.0171380

Braun, V., Clarke, V., 2006. Using thematic analysis in psychology. Qual. Res. Psychol. 3, 77–101. https://doi.org/10.1191/1478088706qp063oa

Chomyn, O., Wapenaar, W., Richens, I.F., Reyneke, R.A., Shortall, O., Kaler, J., Brennan, M.L., 2023. Assessment of a joint farmer-veterinarian discussion about biosecurity using novel social interaction analyses. Prev. Vet. Med. 212, 105831. https://doi.org/10.1016/j.prevetmed.2022.105831

Faltermaier, T., 1997. Why public health research needs qualitative approaches: Subjects and methods in change. Eur. J. Public Health 7, 357–363. https://doi.org/10.1093/eurpub/7.4.357

Hennessey, M., Barnett, T., 2023. Method in limbo? Theoretical and empirical considerations in using thematic analysis by veterinary and One Health researchers. Prev. Vet. Med. 221, 106061. https://doi.org/10.1016/j.prevetmed.2023.106061

Ma, R.S.B., 2000. The role of qualitative research in broadening the ‘evidence base’ for clinical practice. J. Eval. Clin. Pract. 6, 155–163. https://doi.org/10.1046/j.1365-2753.2000.00213.x

O’Brien, B.C., Harris, I.B., Beckman, T.J., Reed, D.A., Cook, D.A., 2014. Standards for Reporting Qualitative Research: A Synthesis of Recommendations. Acad. Med. 89, 1245. https://doi.org/10.1097/ACM.0000000000000388

Reviewer 2:

Comment: An interesting study, I hope this manuscript will be published and read by all poultry stakeholders. Please see on the manuscript for specific comment, however a revision is needed. And if any additional data or information from university/research institution expert, will be broaden the manuscript discussion.

Authors’ response: In this study, universities or research institutions were considered as stakeholders, and therefore were not involved as external experts in the discussion. 

Line 167: The sentence has been reformulated to clarify the meaning. 

Line 341: The headline of this paragraph has been reformulated to better reflect its content. Indeed, this paragraph reflects private stakeholders’ opinion that the academic sector is disconnected from the field realities, which was not corroborated by interviewees from academia.

---

## [Editor Report · Decision Letter 1]

23 Jul 2024

Needs and capabilities for improving poultry production and health management in Indonesia

PONE-D-23-43241R1

Dear Dr. Chapot,

We’re pleased to inform you that your manuscript has been judged scientifically suitable for publication and will be formally accepted for publication once it meets all outstanding technical requirements.

Kind regards,

Biswajit Pal, M.SC., Ph.D

Academic Editor

PLOS ONE
---

## [Editor Report · Acceptance letter]

13 Aug 2024

PONE-D-23-43241R1 

PLOS ONE

Dear Dr. Chapot, 

I'm pleased to inform you that your manuscript has been deemed suitable for publication in PLOS ONE. Congratulations! Your manuscript is now being handed over to our production team.

Kind regards, 

on behalf of

Dr. Biswajit Pal 

Academic Editor

PLOS ONE